# The Development of Population-Specific Spirometric Reference Equations for Iraqi Adults

**DOI:** 10.3390/healthcare13111254

**Published:** 2025-05-26

**Authors:** Alaa Alsajri, Walid Al-Qerem, Dzul Azri Mohamed Noor, Judith Eberhardt

**Affiliations:** 1School of Pharmaceutical Sciences, University Sains Malaysia, Gelugor 11800, Malaysia; alaa94@student.usm.my; 2Department of Pharmacy, Al-Zytoonah University of Jordan, Amman 11733, Jordan; 3Department of Psychology, School of Social Sciences, Humanities and Law, Teesside University, Borough Road, Middlesbrough TS1 3BX, UK

**Keywords:** spirometry, pulmonary disorders, asthma, spirometry equation, COPD, Iraqi adults

## Abstract

**Background**: Spirometry is one of the most important medical tests used to diagnose and monitor diseases affecting the respiratory system. There are several reference equations that can provide reference values to be referred to during the spirometer examination process. However, the equations of the Global Lung Function Initiative 2012 (GLI 2012) Committee were found not to apply to Iraqi populations. Furthermore, there is currently no equation derived from normal values specific to Iraqis. **Objectives**: The aim of this study was to establish a spirometric reference equation that can be used in the Iraqi population to diagnose and monitor respiratory diseases. **Methods**: Spirometry data were collected from healthy, non-smoking Iraqi adults. Generalized additive models of location, scale, and shape were used to construct the equations. Spirometry parameters, including predicted values, predicted percent, and Z-scores, were calculated based on the equations developed. A new spirometry dataset consisting of 344 participants, who were not involved in creating the equations, was used for equation validation. **Results**: The equations were constructed using spirometer data from 966 adults (33.1% women). The equations were evaluated by calculating Z-scores and predicted values, and the results showed that Z-scores for all variables calculated based on the present equation were less than ±0.5. **Conclusions**: To reduce the effect of ethnic and race on the normal values of spirometry parameters, the equations developed in this study can be used to evaluate and diagnose respiratory diseases in Iraqis.

## 1. Introduction

Spirometry is a non-invasive test that is essential for the management and diagnosis of many respiratory diseases [1]. The correct use of spirometers enables accurate diagnosis and facilitates specialized therapeutic interventions by healthcare providers. The interpretation of the results of the spirometry test relies on comparing the values of the different spirometric parameters with the predicted reference values [2]. Therefore, choosing the correct spirometric reference equation is crucial for the success of the test, as normal values vary between individuals depending on age, height, weight, race and other environmental factors [3]. This is crucial to improve the management of respiratory disease including asthma and COPD that have significant burden globally and in Iraq [4]. To reduce the error rate and ensure the appropriate selection of spirometric reference equations, the American Thoracic Society and the European Respiratory Society have approved the use of the Global Lung Function Initiative (GLI) 2012 equations for examining and evaluating lung function globally. The Global Lung Function Initiative 2012 equations are a comprehensive set of equations designed to provide uniform reference values for evaluating lung function across various ethnicities worldwide. The data used to develop the GLI equations were collected from 33 countries and included more than 74,000 participants. However, these equations lack data from the Arab world [5]. The suitability of these equations has been evaluated in several Arab and Asian countries and found to be inappropriate, prompting researchers to develop local versions for calculating lung function [6,7,8,9]. In Iraq, a study was conducted to determine the suitability of the GLI equations for the Iraqi population, revealing that they are inappropriate. The findings highlighted the need to establish a reference equation specifically for examining, diagnosing and monitoring respiratory diseases in Iraqis [10]. The present study aimed to establish a spirometric reference equation for evaluation lung function in adult Iraqis.

## 2. Materials and Methods

The present study was divided into two parts: the first part focused on creating the spirometric reference equations, while the second part evaluated the suitability of these equations for the Iraqi population. Ethical approval for the study was obtained from the Iraqi Ministry of Health (approval number 7/2022).

### 2.1. Participants

Participants were recruited through friends and various social media platforms, primarily Facebook, which is the most popular in Iraq [11]. The health status and smoking habits of the participants were assessed using a questionnaire, which included inquiries such as: “Are you currently a smoker?”, “Do you reside with someone who is smoker?”, “Are you a former smoker?”, and “Do you suffer from any chronic diseases?”. All Iraqis over the age of 18, regardless of sex, were eligible to participate, provided they did not meet any of the exclusion criteria. These criteria included the following: smokers, individuals living with smokers, those with chronic respiratory diseases such as asthma, or non-respiratory conditions such as hypertension, as well as individuals who had recently undergone surgery or those with acute respiratory symptoms. Pregnant women were excluded from this study because hormonal changes during pregnancy are known to affect respiratory system function, as demonstrated in previous studies [12,13].

The study was conducted across various regions of Iraq, including the northern, central, and southern regions. Data were collected from various locations, including markets, stadiums, colleges, hospitals, and pharmacies, after securing the necessary approvals from the relevant authorities. Individuals were informed about the primary objective of the study and asked to take part. After explaining the study and potential risks of participating, such as nausea and dizziness, written consent was obtained from participants. The inclusion and exclusion criteria were determined based on previous studies [14,15].

### 2.2. Measurement and Materials

Anthropometric and Lung function examinations were performed for the participants following the guidelines of the American Thoracic Society and the European Respiratory Society [16]. Participants’ heights were measured using a measuring tape attached to a wall, ensuring accurate readings in meters. The MIR Minispir device was used to perform lung function examinations for all participants. This device is known for its accuracy and ease of use, being a lightweight and portable instrument that connects to a computer. It is widely used in lung function research due to its simplicity and reliability [17]. It was developed by Medical International Research in Rome, Italy [18]. The equipment complies with ATS/ERS standards for spirometry [16]. It was operated using WinspiroPRO software (version 8.5) and has a volume accuracy of ±3% or 50 mL, and a flow accuracy of ±5% or 200 mL/s. The device has been employed in numerous previous studies to assess lung function [9,19].

### 2.3. Statistical Analysis

The data were analyzed using the Statistical Package for the Social Sciences (SPSS), version 23. To determine whether the anthropometric and respiratory data were normally distributed between the two sexes, the Kolmogorov–Smirnov was used, revealing that the data were not normally distributed. Generalized Additive Models for Location, Scale, and Shape (GAMLSS) is a versatile statistical method used for modeling and predicting the dispersion of a response variable. Unlike conventional regression models, which focus primarily on estimating the mean of the response variable, GAMLSS allows for the modeling of multiple distribution parameters, including location (mean), scale (variance), and shape (skewness and kurtosis) [20]. This method allows for the application of different link functions and data distributions, including the Box-Cox-Cole-Green (BCCG) distribution, normal distribution, and Box-Cox power exponential (BCPE). Additionally, smoothing splines can be incorporated into equations to improve data fit. In this study, various models were evaluated for each spirometric value by sex, and the best model was selected based on degrees of freedom (df) and the Schwarz Bayesian criterion (SBC). The models that produced the lowest SBC and df, while exhibiting an appropriate distribution of residuals, were deemed the most suitable for the data.

To determine whether the developed equations were suitable for application to Iraqi populations, data were collected from 344 healthy participants (164 males and 180 females) who were not included in the first phase of equation creation. This sample size exceeded the minimum number recommended by the GLI 2012 guidelines for validating spirometric reference equations in a given population (300 participants, with at least 150 females and 150 males) [21]. The predicted values and Z-scores for FEV_1_, FVC, FEF_25–75%_, and FEV_1_/FVC% were calculated for each participant using the present developed reference equations (referred to as Iraqi Spirometric Equation (ISE)) and different GLI equations including GLI other/mixed (GLI-O), GLI race-neutral (GLI-N), and (GLI Caucasians) GLI-C. The Mann–Whitney U test was applied to evaluate differences in spirometric parameters between the equation and validation data.

## 3. Results

### 3.1. Phase One: Formulating an Equation

Data from a total of 966 participants were analyzed, comprising 646 males and 320 females, out of an initial pool of 2060 participants. A total of 1094 individuals were excluded because they did not meet inclusion criteria (Figure 1).

As seen in Table 1, the pulmonary parameters (FVC, FEV_1_, and FEF_25–75_) were higher in males than in females, except for FEV_1_/FVC%, which was higher in females. Table A1 presents the models studied. Based on the SBC values and degrees of freedom (df), the models were compared, and the best model was selected to formulate the spirometric equations. The SBC values and degrees of freedom for the studied models are presented in Table A2 (males) and Table A3 (females). Spirometric equations were developed for both sexes and are presented in Table 2 (males) and Table 3 (females).

#### 3.1.1. Spirometry Indices in Males

The majority of the data were concentrated within the younger age intervals, resulting in a positively skewed (right-skewed) distribution (Figure A1). FEV_1_ values change with age, increasing until about 26 years, after which they decreased with advancing age. Although the relationship between FEV_1_ and age was clearly inverse, applying a logarithmic transformation improved the linearity of the relationship (Figure A2). Applying a logarithmic transformation also improved the linearity of this relationship between height and FEV_1_ (Figure A3). Model 3 was selected to produce the predicted equation for FEV_1_ because it had the lowest SBC and an appropriate df compared to the other models (Table A1). Residual analysis was conducted to ensure that the chosen model was appropriate for the study data and did not overlook any influencing variables. The analysis demonstrated that the selected model accurately represented the data, with only minor discrepancies that did not indicate any significant issues.

The distribution of the residuals appeared balanced, which is a positive aspect. Most data points aligned with the reference line, suggesting that the residuals are approximately normally distributed. Although some irregularities were observed in the tails, these were typical and remained within acceptable limits (Figure A4). The Worm Plots depicted a favorable model fit, with deviations centered around zero in most panels. While some deviations were present, they were generally within acceptable limits. The symmetry around zero further indicated that the residuals were fairly normally distributed (Figure A5).

For FVC, after reviewing the different models, Model 2 was found to be the best fit for the study data. While Model 1 had a slightly lower SBC, its higher degrees of freedom made Model 2 the more suitable choice. Additionally, Model 2 exhibited good normality of residuals, favorable Worm Plots, and consistent centiles. After evaluating seven models to determine the most suitable for FEV_1_/FVC in men, Model 1 was identified as the best, as it had the lowest SBC among the other models.

Although Model 1 had the lowest SBC in FEF_25–75%_, indicating it might be the best fit, Model 2 was very close in performance and exhibited better df, making it the preferred model for the present data. Model 2 also exhibited good normality of residuals, favorable Worm Plots, and consistent centiles compared to Model 1.

#### 3.1.2. Spirometry Parameters in Females

The female sample included in the study was negatively skewed towards the younger ages (Figure A6). Upon evaluating the GAMLSS models, Model 1 was identified as the best fit for calculating FEV_1_. Although both models had very similar values in terms of df and SBC (with Model 1 exhibiting a slightly lower SBC), Model 1 was superior in terms of residuals, which had a rate of −0.002, closer to zero than that of Model 2.

After evaluating different models for the FVC data for females, Model 1 was identified as the most suitable for the data, compared to the rest of the models, based on its SBC and df, which were lower compared to other models. The residuals of Model 1, along with its variance, skewness, and kurtosis, indicated a good fit for the data. In FEV_1_/FVC, Model 1 was identified as the most suitable based on its SBC and df, which were lower compared to the other models. Finally, in FEF_25–75%_, Model 1 was selected due to its lowest standardized SBC and acceptable df. Additionally, it demonstrated well-formed residual normality, making it the most suitable choice.

### 3.2. Equation Validation

The anthropometric and spirometric parameters for the participants in the second phase of the study are displayed in Table 4. Moreover, the Mann–Whitney U test indicated that there were no significant differences between the spirometric values of the samples used in phase one and phase two, except for FEF_25–75%_ in males (*p* = 0.01) and FEV_1_/FVC% in females (*p* = 0.01).

The results of the validation study were compared against those of the GLI-O, GLI-C, and GLI-N equations, as shown in Table 5 and Table 6. For males, the results show that the ISE produced the best fit for all spirometric values. In FEV_1_, the ISE provided the Z-score closest to zero (0.06), indicating the best agreement with the observed FEV_1_ values. The GLI-C equation slightly overestimated FEV_1_, while the GLI-O and GLI-N equations slightly underestimated it. In FVC, similar to FEV1, the ISE showed the best alignment with the observed FVC values (Z-score = 0.08). The GLI-C equation overestimated FVC, while the GLI-O and GLI-N equations showed slight underestimations. In FEV1/FVC%, the ISE again provided the Z-score closest to zero (−0.05), indicating the most accurate prediction of the FEV_1_/FVC ratio. All GLI-based equations slightly underestimated the ratio. For FEF_25–75%_, the ISE produced the most accurate predictions, with a Z-score closest to zero (0.10). The GLI-C and GLI-O equations slightly underestimated the flow, with GLI-O demonstrating the largest deviation. Moreover, the standard deviation was closest to 1, and the predicted percent was closest to 100% when using the ISE.

The results for females indicate that the ISE consistently generates Z-scores closest to zero across all spirometry parameters (Table 7), making it the most accurate and suitable equation for the Iraqi female population. However, for FEV_1_, the GLI-C equation had the lowest Z-score. Nonetheless, the ISE retained the closest standard deviation to 1 among all the studied equations for FEV_1_. The GLI-C equation tended to overestimate FEV1 and FVC but underestimated the FEV_1_/FVC ratio and FEF_25–75%_. Conversely, the GLI-O and GLI-N equations slightly underestimated FEV1 and FVC, while showing greater deviations in FEF_25–75%_. The ISE consistently provided a predicted percent closest to 100%, making it the most suitable equation for the Iraqi female population.

## 4. Discussion

Iraqi society lacks reference equations for calculating and evaluating lung function, as the GLI 2012 equations were found to be unsuitable for this population and may yield inaccurate results, potentially affecting the diagnosis and evaluation of lung conditions in Iraqis [22].

The GLI-2012 equations were created to establish globally applicable predictive formulas for spirometric reference values across broad age groups. However, limitations in these equations may restrict their utility and highlight the need for population-specific models tailored to distinct demographics. A key shortcoming is the GLI-2012’s failure to account for intra-ethnic diversity. For example, the GLI-C category pooled data from Europe, the Middle East (e.g., Israel), and North Africa, regions with significant demographic heterogeneity, raising concerns about the equations’ generalizability. Additionally, the GLI-2012 relies on data collected from 1978 onward, despite evidence of a global upward trend in average height over time [22], which may influence lung function metrics. A Japanese study highlighted how shifting anthropometrics can alter spirometric results. Furthermore, ethnic groups residing in different geographic regions exhibit variations in lung function, as seen in comparisons between Indian individuals born in the United States and those in India, with the former showing higher spirometric values even after adjusting for height and age [23]. These findings emphasize the necessity of refining predictive models to reflect evolving and region-specific population characteristics.

Several countries in the Middle East have recognised that globally derived spirometric reference values are not fully applicable to their populations, leading to the development of population-specific equations. Reference values established for adults in the region highlight the significant influence of ethnicity and geographic factors on lung function [24]. Similarly, spirometric norms developed for healthy adults in Jordan, Oman, and Saudi Arabia differ from Western and global equations, reinforcing the need for region-specific standards in clinical practice [6,24,25].

To address this issue for Iraqis, the present study developed population-specific reference equations to examine and monitor lung function. These equations were derived by analyzing spirometry data from 966 Iraqi adults and applying statistical methods to formulate the reference equations. The GAMLESS approach was used to construct the equations because it effectively handle non-linear data, overcoming the limitations of traditional models [26,27]. The LMS method, commonly used in growth and lung function modeling, captures the median (M), variability (S), and skewness (L) through the BCCG distribution. However, it is a special case within the broader GAMLSS framework, which enables more flexible modeling by allowing separate smooth functions for location, scale, skewness, and kurtosis [28]. This makes GAMLSS particularly suitable for data that are non-normal and heteroscedastic, as is often the case in spirometry across age groups [26]. For example, Lung volume increases until around the age of 30 due to natural growth and increased muscle mass. After this point, it begins to gradually decline, particularly in individuals over 30, due to muscle weakness, reduced lung elasticity, and frequent exposure to pollutants [29]. While recent studies in Japan and Korea [30,31,32] have utilized the LMS method, they were focused on homogeneous populations. For heterogeneous populations, studies like those from the GLI Network have used GAMLSS to develop robust multi-ethnic reference equations for different ethnicities [5,24].

To test the applicability of the spirometry reference equations developed in this study for the Iraqi population, data of 344 healthy, non-smoking Iraqi individuals (52.3% females) who were not included in the equation creation phase were analyzed. To evaluate suitability, the Z-score was calculated for all spirometric variables in this study based on the newly developed equations. Ideally, Z-scores should be normally distributed, with a mean close to zero and a standard deviation close to 1. However, achieving a mean Z-score of exactly zero is challenging due to physiological differences among individuals within the same environment, race, and society. To account for these variations, the GLI 2012 Committee adopted a Z-score cut-off point of less than ±0.5 instead of zero [5]. Z-scores are superior to predicted % as they are independent of changes in height, age, and race [26] and were therefore adopted to determine the suitability of the equations in the second phase of this study. In the present study, the Z-scores for all variables were less than ±0.5, confirming that the equations are suitable for application in the Iraqi population. Moreover, when comparing the Z-scores produced by the ISE with those from different GLI equations, the ISE produced the Z-scores closest to zero across all studied spirometric parameters in both sexes, except for FEV_1_ in females. Furthermore, many of the GLI equations exceeded the critical Z-score cut-off point, including GLI-O in FEV_1_ and FEF_25–75%_ in males, as well as FEV_1_ and FEV_1_/FVC% in females. Additionally, GLI-C exceeded the cut-off point in FEV_1_/FVC% in females. These results confirm the superiority of ethnic-specific equations over ethnically diverse ones, a finding supported by several previous studies [6,7,8,9]. While international guidelines like GINA provide general spirometric reference values, national guidelines often use population-specific equations that consider local factors such as ethnicity, height, and weight. These locally derived equations can improve the accuracy of diagnosing and classifying conditions like asthma and COPD. For example, the KNHANES-VI equation used in South Korea showed greater sensitivity in detecting airway obstruction than the globally used GLI-2012 equation [33,34]. The integration of these reference values into national COPD and asthma guidelines would create a more precise foundation on which to base disease staging and treatment planning in patients from Iraq, ensuring that it is based on population-specific norms rather than on the possibly mismatched international equations.

### 4.1. Strengths and Limitations

This study is the first to provide a spirometric reference equation for Iraqi adults in accordance with the recommendations of the American Thoracic Society, the European Respiratory Society, and the GLI 2012 committee. This marks an important step in addressing the lack of population-specific reference equations for lung function in Iraq. The equations developed here provide a foundation for further validation through their application to patients with respiratory conditions, particularly those with COPD.

Despite its strengths, this study has several limitations that warrant discussion and suggest avenues for future research. First, it did not account for environmental, physical, biological, and nutritional factors that may affect respiratory rates and lung volume [35]. Future research should include these variables to evaluate their impact on spirometric parameters and to refine the equations accordingly. Although the present study did not specifically assess several risk factors, such as occupational exposure and biomass smoke exposure, it excluded participants with acute or chronic respiratory symptoms. Moreover, the study followed the selection criteria used in global and regional spirometric reference studies, including the GLI study [5], which also did not account for these exposures. While this limitation may have a minor impact on the study’s findings, future research should consider evaluating these exposures more closely to further refine population-specific reference standards.

Another limitation may be the young age of the study sample (in both phases), which had a median age of 32.5 years for females and 27.3 years for males. However, this is generally consistent with the demographics of the Iraqi population, which has a median age of 22.4 years [36]. Moreover, previously published equations have reported comparable or younger median ages [6,37,38]. Nevertheless, 187 females and 187 males in the total sample were aged 40 years or older. Expanding future studies to include older age groups would further enhance the reliability of the produced equations.

Finally, the internet-based recruitment method used in this study may have introduced selection bias, as individuals without internet access were less likely to be enrolled. Nevertheless, with the global rise in internet connectivity, the recent literature suggests that the socio-demographic characteristics of participants recruited online increasingly resemble those of the general population [39]. This is particularly relevant for Iraq, where internet access has reached approximately 81.7% as of early 2025 [40]. Moreover, excluding individuals under the age of 18 may further increase this proportion.

### 4.2. Future Directions

The application of population-specific reference values for adults in Iraq will improve the accuracy of spirometry interpretation, reducing the risk of underdiagnosing or misdiagnosing obstructive and restrictive lung diseases. This, in turn, will support clinicians in making more reliable treatment decisions based on the characteristics of the local population. Additionally, these reference equations may strengthen respiratory screening programmes in Iraq by improving the early detection of airway obstruction, particularly among asymptomatic individuals and those in rural or disadvantaged communities.

## 5. Conclusions

This study developed spirometric reference equations specifically for adult Iraqis, addressing a critical gap in population-specific tools for evaluating lung function. The suitability and applicability of these equations were validated, with Z-scores demonstrating a mean close to zero and a standard deviation near one, indicating that the equations are robust for use in this population.

The findings of this study support the application of these equations to larger samples of healthy Iraqis to further refine their accuracy and generalizability. Additionally, future research should focus on applying these equations to patients with chronic respiratory diseases, such as COPD, to assess their reliability in diagnosing and monitoring respiratory conditions. Expanding their use to such populations will help determine their potential for integration into clinical practice for managing chronic respiratory diseases in Iraq.

By providing the first spirometric reference equations tailored to the Iraqi population, this study represents a significant advancement in respiratory health assessment and paves the way for further research to enhance their applicability and clinical relevance.

## Figures and Tables

**Figure 1 healthcare-13-01254-f001:**
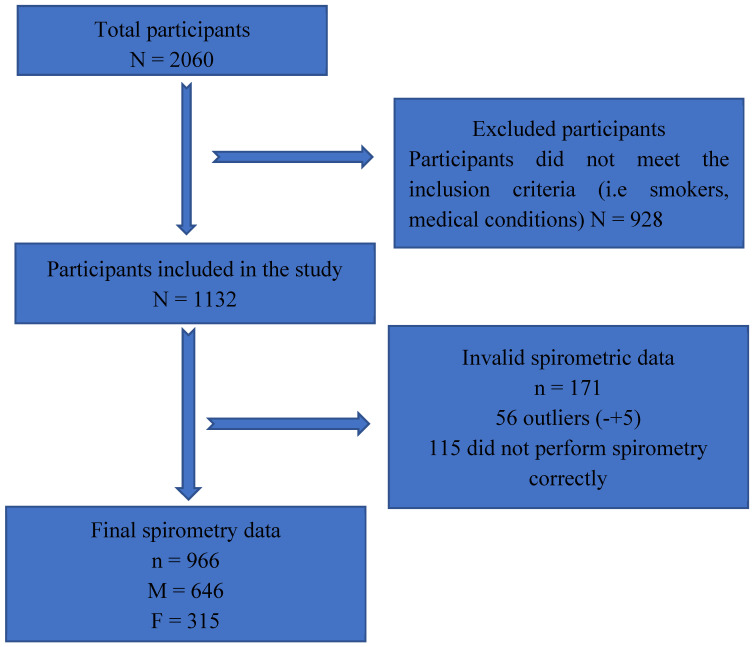
Flowchart of the participant recruitment and data collection process.

**Table 1 healthcare-13-01254-t001:** Descriptive analysis of males and females.

Parameters	Males		Females	
	Median (Q1–Q3)	Range	Median (Q1–Q3)	Range
Height (cm)	174.0 (170.0–178.0)	155–193	157 (154–161)	154–176
Age (year)	28.57 (22.72–40.68)	18–70	34.79 (24.33–45.92)	18–67
FEV_1_ (Liter)	3.98 (3.67–4.42)	2.42–6.23	2.91 (2.63–3.19)	1.70–5.04
FVC (Liter)	4.64 (4.22–5.17)	2.68–7.78	3.18 (2.85–3.63)	1.92–6.24
FEV_1_/FVC%	0.86 (0.83–0.89)	0.72–1.00	0.90 (0.85–0.96)	0.71–1.00
FEF_25–75%_ (L/S)	4.44 (3.94–5.03)	2.31–8.52	3.55 (3.00–4.08)	1.77–6.11

Abbreviations: FEF_25–75_, forced expiratory flow between 25% and 75% of forced vital capacity; FEV_1_, forced expiratory volume in the first second; FEV_1_/FVC, forced expiratory volume in the first second/forced vital capacity; FVC, forced vital capacity; Q1, first quantile, Q3 third quantile.

**Table 2 healthcare-13-01254-t002:** Spirometric reference equations for males.

FEV_1_	Mean Sigma Nu	−28.1570 + (6.8055 × log(height)) − (0.8566 × log(age)) + Mspline (age) exp (−1.8308 − (0.1192 × log(age)) + Sspline (age)) − 2.3096 + (0.3956 × log(age))
FVC	Mean sigma nu	exp (−8.37157 + (2.05757 × log(height)) − (0.20888 × log(age)) exp (−2.14143) − 1.8044
FEV_1_/FVC	Mean sigma nu	1.458485 − (0.111511 × log(height) − (0.007171 × log(age))+ Mspline (age) exp ((−3.39125 + 0.15846 × (log(age)) + Ssplie (age)) 0.4028 − (0.4593 × log(age))
FEF_25–75_	Mean sigma nu	exp ((−4.20476 + (1.23963 × log(height) − (0.20600 × log(age)) + Mspline (age)) exp (−2.42013 + (0.18673 × log(age)) + Sspline (age)) −4.8567 + 1.0254 × log(age)

FEF_25–75_, forced expiratory flow between 25% and 75% of forced vital capacity; FEV_1_, forced expiratory volume in the first second; FEV_1_/FVC, forced expiratory volume in the first second/forced vital capacity; FVC, forced vital capacity. log, logarithm; Exp, Exponential function.

**Table 3 healthcare-13-01254-t003:** Spirometric reference equations for females.

FEV_1_	Mean Sigma Nu	−24.46931 + (5.85935 × log(height)) − (0.65217 × log(age)) + Mspline (age) exp (−2.8156 + (0.1647 × log(age)) + (Sspline (age)) −2.2930 + (0.2341 × log(age))
FVC	Mean sigma nu	exp (−11.26749 + (2.61061 × log(height)) − (0.22545 × log(age)) exp (−2.08730) −1.8645
FEV_1_/FVC	Mean sigma nu	exp (1.968086 − (0.416269 × log(height)) + (0.009565 × log(age))+ Mspline (age) exp (−3.7224 + (0.3145 × (log(age)) + Sspline (age)) 1
FEF_25–75_	Mean sigma nu Tau	exp (−3.5388 + (1.0490 × log(height)) − (0.1505 × log(age)) + Mspline (age)) exp (−3.9378 + (0.6482 × log(age)) + Sspline −7.711 + (1.956 × log(age)) exp (0.6736 + (0.637 × log(age))

FEF_25–75_, forced expiratory flow between 25% and 75% of forced vital capacity; FEV_1_, forced expiratory volume in the first second; FEV_1_/FVC, forced expiratory volume in the first second/forced vital capacity; FVC, forced vital capacity. log, logarithm; Exp, exponential function.

**Table 4 healthcare-13-01254-t004:** Anthropometric parameters of participants in Phase Two.

	Median	Q1–Q3	Minimum	Maximum
Males N = 164 (47.7%)				
Age	23.94	20.96–30.22	18.02	67.46
Height	171.00	168.00–176.00	155	190
FEV_1_	4.08	3.73–4.53	2.43	5.83
FVC	4.81	4.28–5.31	2.68	7.17
FEV_1_/FVC	0.86	0.83–0.89	64	100
FEF_25–75%_	4.64	4.16–5.48	2.46	8.52
Females N = 180 (52.3%)				
Age	29.00	22.07–44.85	18.00	69.9
Height	159.00	154.00–162.00	115	176
FEV_1_	2.90	2.55–3.20	1.51	5.04
FVC	3.29	2.85–3.68	1.69	6.24
FEV_1_/FVC	0.88	0.85–0.93	57.8	100
FEF_25–75%_	3.39	2.94–4.00	1.71	5.32

**Table 5 healthcare-13-01254-t005:** Spirometric parameters of participants in Phase Two. Data presented as median (Q1–Q3).

Parameters	Predicted Value	Predicted Percent	Z-Score
Males N = 164			
FEV_1_	4.10 (3.82–4.36)	102.66% (95.10–107.56)	0.06 (0.99)
FVC	4.67 (4.34–5.04)	103.94% (94.84–110.35)	0.08 (1.07)
FEV_1_/FVC	0.86 (0.85–0.87)	99.36% (96.46–103.56)	−0.05 (1.09)
FEF_25–75%_	4.53 (4.27–4.76)	105.24% (92.21–121.84)	0.10 (1.31)
Females N = 180			
FEV_1_	3.02 (2.65–3.22)	97.19% (88.60–106.45)	−0.31 (1.20)
FVC	3.32 (2.87–3.64)	98.56% (89.70–107.22)	−0.22 (1.25)
FEV_1_/FVC	0.90 (0.89–0.91)	97.95% (93.95–103.38)	−0.12 (0.56)
FEF_25–75%_	3.57 (3.28–3.72)	96.01% (84.94–110.68)	−0.27 (1.20)

Abbreviations: FEF_25–75%_, forced expiratory flow between 25% and 75% of forced vital capacity; FEV_1_, forced expiratory volume in the first second; FEV_1_/FVC, forced expiratory volume in the first second/forced vital capacity; FVC, forced vital capacity; SD, standard deviation; N, number of participants.

**Table 6 healthcare-13-01254-t006:** Spirometric parameters and Z-score of studied equations (for males).

Parameters	Equations	Predicted-V	Predicted-P	Z-Score
FEV_1_	ISE	4.10 (3.82–4.36)	102.66% (95.10–107.56)	0.06 (0.99)
	GLI-C	4.22 (3.95–4.49)	98.54% (91.73–103.94)	−0.10 (0.87)
	GLI-O	3.93 (3.68–4.19)	105.77% (98.46–111.57)	0.52 (0.95)
	GLI-N	3.92 (3.67–4.21)	105.67% (98.66–111.87)	0.47 (0.87)
FVC	ISE	4.67 (4.34–5.04)	103.94% (94.84–110.35)	0.08 (1.07)
	GLI-C	4.98 (4.74–5.32)	94.87% (88.32–103.64)	−0.33 (0.93)
	GLI-O	4.58 (4.37–4.90)	103.03% (95.92–112.55)	0.38 (1.07)
	GLI-N	4.56 (4.27–4.87)	103.25% (96.57–113.39)	0.36 (0.90)
FEV_1_/FVC	ISE	0.86 (0.85–0.87)	99.36% (96.46–103.56)	−0.05 (1.09)
	GLI-C	0.85 (0.83–0.86)	101.57% (98.97–106.37)	0.38 (0.86)
	GLI-O	0.86 (0.84–0.87)	100.49% (97.92–105.25)	0.26 (0.92)
	GLI-N	0.86 (0.84–0.86)	100.61% (98.05–105.45)	0.29 (0.92)
FEF_25–75%_	ISE	4.53 (4.27–4.76)	105.24% (92.21–121.84)	0.10 (1.31)
	GLI-C	4.54 (4.19–4.85)	107.29% (93.85–124.74)	0.29 (0.81)
	GLI-O	4.31 (3.97–4.60)	113.15% (98.96–131.54)	0.51 (0.83)
	GLI-N	NA	NA	NA

N: race neutral, C: Caucasians, O: other/mixed, ISE: Iraqi Spirometric Equation, NA: Not Applicable.

**Table 7 healthcare-13-01254-t007:** Spirometric parameters and Z-score of studied equations (for Females).

Parameters	Equation	Predicted-V	Predicted-P	Z-Score
FEV_1_	ISE	3.02 (2.65–3.22)	97.19% (88.60–106.45)	−0.31 (1.20)
	GLI-C	3.01 (2.65–3.23)	96.61% (89.11–106.85)	−0.06 (1.38)
	GLI-O	2.80 (2.47–3.01)	103.70% (95.65–114.15)	0.53 (1.48)
	GLI-N	2.81 (2.48–3.03)	102.95% (94.66–113.93)	0.44 (1.43)
FVC	ISE	3.32 (2.87–3.64)	98.56% (89.70–107.22)	−0.22 (1.25)
	GLI-C	3.49 (3.18–3.77)	91.91% (82.97–102.02)	−0.41 (1.38)
	GLI-O	3.22 (2.92–3.47)	99.90% (90.18–110.88)	0.25 (1.56)
	GLI-N	3.22 (2.89–3.48)	99.68% (90.72–111.31)	0.22 (1.47)
FEV_1_/FVC	ISE	0.90 (0.89–0.91)	97.95% (93.95–103.38)	−0.12 (0.56)
	GLI-C	0.85 (0.82–0.88)	104.68% (98.93–108.61)	0.69 (1.16)
	GLI-O	0.86 (0.82–0.89)	103.57% (97.89–107.46)	0.57 (1.24)
	GLI-N	0.86 (0.82–0.89)	103.56% (98.10–107.33)	0.57 (1.21)
FEF_25–75%_	ISE	3.57 (3.28–3.72)	96.01% (84.94–110.68)	−0.27 (1.20)
	GLI-C	3.48 (2.82–3.74)	103.51% (90.55–120.26)	0.29 (0.81)
	GLI-O	3.30 (2.68–3.55)	109.15% (95.49–126.82)	0.42 (0.82)
	GLI-N	NA	NA	NA

N: race neutral, C: Caucasians, O: other/mixed, ISE: Iraqi Spirometric Equation, NA: Not Applicable.

## Data Availability

All data studied here are available from H. Al-Sajri, A. (2024). Spirometric reference equation [Data set]. Zenodo. https://doi.org/10.5281/zenodo.14562300. Accessed on 27 December 2024.

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
