# Peer review of "The Development of Population-Specific Spirometric Reference Equations for Iraqi Adults"

_healthcare, 2025, doi:10.3390/healthcare13111254_

Round 1

Reviewer 1 Report

Comments and Suggestions for Authors

This study attempts to develop population-specific spirometric reference equations for Iraqi adults.

The topic is original, relevant and addresses the lack of spirometric reference for the Iraqi population.

Conclusions are consistent with the data presented in the results section.

Below are a few suggestions:

Recruitment through social media may introduce selection bias unlike random sampling. Consider mentioning this in the discussion section.

There does not seem to be any data regarding factors that can affect spirometry such as occupational exposures. Consider further discussing about this.

Line 138 under Results - correct “date” to data

Author Response

This study attempts to develop population-specific spirometric reference equations for Iraqi adults.

The topic is original, relevant and addresses the lack of spirometric reference for the Iraqi population.

Conclusions are consistent with the data presented in the results section.

Response: We thank Reviewer 1 for their positive evaluation of our manuscript and constructive suggestions. We address each point below.

Below are a few suggestions:

[Comment] Recruitment through social media may introduce selection bias unlike random sampling. Consider mentioning this in the discussion section.

-Response: Thank you for your comment. The following has been added to the limitations section: “Finally, the internet-based recruitment method used in this study may have introduced selection bias, as individuals without internet access were less likely to be enrolled. Nevertheless, with the global rise in internet connectivity, recent literature suggests that the socio-demographic characteristics of participants recruited online increasingly resemble those of the general population [39]. This is particularly relevant for Iraq, where inter-net access has reached approximately 81.7% as of early 2025 [40]. Moreover, excluding individuals under the age of 18, may further increase this proportion.”

[Comment]  There does not seem to be any data regarding factors that can affect spirometry such as occupational exposures. Consider further discussing about this.

-Response: The following has been added to the limitations section: “Although the present study did not specifically assess several risk factors, such as occupational exposure and biomass smoke exposure, it excluded participants with acute or chronic respiratory symptoms. Moreover, the study followed the selection criteria used in global and regional spirometric reference studies, including the GLI study [5], which also did not account for these exposures. While this limitation may have a minor impact on the study’s findings, future research should consider evaluating these exposures more closely to further refine population-specific reference standards.”

[Comment]  Line 138 under Results - correct “date” to data

The typographical error has been corrected as advised.

Reviewer 2 Report

Comments and Suggestions for Authors

Thank you for inviting me to review this paper on hot topic!

I have several proposals:

  1. please exclude this phrase from results section:
    This section may be divided by subheadings. It should provide a concise and precise description of the experimental results, their interpretation, as well as the experimental conclusions that can be drawn.
  2. limitations must be completed: inclusion of patients via social media etc
  3. please add future directions: how implementation of reference values will change real life clinical practice in Iraq
  4. what impact will be on local guidelines of COPD and asthma in Iraq?
  5. what impact will be on local screening programs in Iraq?
  6. please in discussion add information about references values from countries from region!

Author Response

Thank you for inviting me to review this paper on hot topic!

We thank Reviewer 2 for the positive assessment of our work and for the helpful suggestions, which have strengthened the clinical relevance of our manuscript.

I have several proposals:

  1. [Comment] please exclude this phrase from results section:
    This section may be divided by subheadings. It should provide a concise and precise description of the experimental results, their interpretation, as well as the experimental conclusions that can be drawn.

Response: We thank the reviewer for noticing this. The placeholder text has been removed from the Results section.

2. [Comment] limitations must be completed: inclusion of patients via social media etc

Response: The limitation section has been thoroughly expanded and the following has been added: “Finally, the internet-based recruitment method used in this study may have introduced selection bias, as individuals without internet access were less likely to be enrolled. Nevertheless, with the global rise in internet connectivity, recent literature suggests that the socio-demographic characteristics of participants recruited online increasingly resemble those of the general population [39]. This is particularly relevant for Iraq, where inter-net access has reached approximately 81.7% as of early 2025 [40]. Moreover, excluding individuals under the age of 18, may further increase this proportion.”

3. [Comment] please add future directions: how implementation of reference values will change real life clinical practice in Iraq

4. what impact will be on local guidelines of COPD and asthma in Iraq?

5. what impact will be on local screening programs in Iraq?

-Response (3-5): A future directions section has been added, which includes the following:” The application of population-specific reference values for adults in Iraq will improve the accuracy of spirometry interpretation, reducing the risk of underdiagnosing or misdiagnosing obstructive and restrictive lung diseases. This, in turn, will support clinicians in making more reliable treatment decisions based on the characteristics of the local population. Additionally, these reference equations may strengthen respiratory screening programmes in Iraq by improving the early detection of airway obstruction, particularly among asymptomatic individuals and those in rural or disadvantaged communities.”

6. [Comment]  please in discussion add information about references values from countries from region!

  Response: the following was added: “Several countries in the Middle East have recognised that globally derived spirometric reference values are not fully applicable to their populations, leading to the development of population-specific equations. Reference values established for adults in the region highlight the significant influence of ethnicity and geographic factors on lung function [24]. Similarly, spirometric norms developed for healthy adults in Jordan, Oman, and Saudi Arabia differ from Western and global equations, reinforcing the need for region-specific standards in clinical practice [6,24,25].”

Reviewer 3 Report

Comments and Suggestions for Authors

Thank you for the opportunity to review your manuscript, which aimed to establish a spirometric reference equation for diagnosing and monitoring respiratory diseases in the Iraqi population.
I have some comments for you before considering a final decision on its suitability.

In the Participants section, you describe different questions regarding tobacco smoking in order to exclude tobacco smokers, but you do not address exposure to biomass-burning smoke, which is a major cause of lung obstruction and is even more common than tobacco smoking. Please comment on this matter.

Methods section: Detail the specifications entered for the MIR Minispir device.

The use of "eponymous," for example, in Al-Sajari, is not recommended. 
The lung function parameters in Tables 1, 4, and 5 should include percentages.
Patients should be stratified into age groups, in decades.

Minor comments: 
There is a double point in line 108.
In the FEV1 acronym, the number 1 should be in subscript.
Remove the paragraph that explains how to fill out the Word format at the beginning of the Results section. Also, correct the footnote for Figure 1.
Correct "quantile" in the footnote for Figure 1.
Table 5 is repeated.

Author Response

[Comment] Thank you for the opportunity to review your manuscript, which aimed to establish a spirometric reference equation for diagnosing and monitoring respiratory diseases in the Iraqi population.
I have some comments for you before considering a final decision on its suitability.

Response: We thank Reviewer 1 for their positive evaluation of our manuscript and constructive suggestions. We address each point below

[Comment] In the Participants section, you describe different questions regarding tobacco smoking in order to exclude tobacco smokers, but you do not address exposure to biomass-burning smoke, which is a major cause of lung obstruction and is even more common than tobacco smoking. Please comment on this matter.

-Response: We thank the reviewer for raising this important point. Please note that Iraq experiences a high prevalence of tobacco smoking, reaching up to 20%, particularly among males:

https://www.emro.who.int/irq/iraq-infocus/world-no-tobacco-day.html.

In contrast, one study reported that the contribution of biomass-burning exposure in Iraq was minimal:

https://iopscience.iop.org/article/10.1088/1755-1315/1262/2/022016.

Nevertheless, the following was added to the limitations section: “Although the present study did not specifically assess several risk factors, such as occupational exposure and biomass smoke exposure, it excluded participants with acute or chronic respiratory symptoms. Moreover, the study followed the selection criteria used in global and regional spirometric reference studies, including the GLI study [5], which also did not account for these exposures. While this limitation may have a minor impact on the study’s findings, future research should consider evaluating these exposures more closely to further refine population-specific reference standards.”

[Comment] Methods section: Detail the specifications entered for the MIR Minispir device.

Response: The following was added:” The MIR Minispir device was used to perform lung function examinations for all participants. This device is known for its accuracy and ease of use, being a lightweight and portable instrument that connects to a computer. It is widely used in lung function research due to its simplicity and reliability [17]. It was developed by Medical International Research in Rome, Italy [18].  The equipment complies with ATS/ERS standards for spirometry [16]. It was operated using WinspiroPRO software (version 8.5) and has a volume accuracy of ±3% or 50 mL, and a flow accuracy of ±5% or 200 mL/s. The device has been employed in numerous previous studies to assess lung function [9,19].”

The use of "eponymous," for example, in Al-Sajari, is not recommended. 

-Response: We agree and have changed the equation name. The equation is now referred to as the Iraqi Spirometric Equation (ISE) to ensure clarity and objectivity.

[Comment] The lung function parameters in Tables 1, 4, and 5 should include percentages.

Response: Thank you for your comment. Tables 1 and 4 present observed values. However, percentage-predicted values for FEV₁, FVC, and FEF₂₅–₇₅% have been added in Tables 5 and 6, based on the developed reference equations.
[Comment]  Patients should be stratified into age groups, in decades.

Response: We thank the reviewer for the suggestion. Appendix Figures A1 and A6 present the age distribution of participants, stratified by 5-years bin, providing a clear overview of age representation within the study sample.

Minor comments: 
There is a double point in line 108.
In the FEV1 acronym, the number 1 should be in subscript.
Remove the paragraph that explains how to fill out the Word format at the beginning of the Results section. Also, correct the footnote for Figure 1.
Correct "quantile" in the footnote for Figure 1.
Table 5 is repeated.

-Response: Thank you for your comment. All of these minor revisions have been performed.

Round 2

Reviewer 2 Report

Comments and Suggestions for Authors

accept as it

Reviewer 3 Report

Comments and Suggestions for Authors

Thank you for addressing my previous concerns.